# Mutation of Signal Transducer and Activator of Transcription 5 (STAT5) Binding Sites Decreases Milk Allergen α_S1_-Casein Content in Goat Mammary Epithelial Cells

**DOI:** 10.3390/foods11030346

**Published:** 2022-01-26

**Authors:** Ning Song, Jun Luo, Lian Huang, Saige Zang, Qiuya He, Jiao Wu, Jiangtao Huang

**Affiliations:** 1Shaanxi Key Laboratory of Molecular Biology for Agriculture, College of Animal Science and Technology, Northwest A&F University, Yangling 712100, China; songning@nwafu.edu.cn (N.S.); kin8248806@163.com (L.H.); zsg0304@126.com (S.Z.); heqiuya826@163.com (Q.H.); wujiao2019@nwafu.edu.cn (J.W.); jiangtaoh@nwafu.edu.cn (J.H.); 2College of Animal Science and Technology, Anhui Agricultural University, Hefei 230036, China

**Keywords:** α_S1_-casein, *CSN1S1* promoter, STAT5, milk allergy, goat mammary epithelial cells

## Abstract

α_S1_-Casein (encoded by the *CSN1S1* gene) is associated with food allergy more than other milk protein components. Milk allergy caused by α_S1_-casein is derived from cow milk, goat milk and other ruminant milk. However, little is known about the transcription regulation of α_S1_-casein synthesis in dairy goats. This study aimed to investigate the regulatory roles of signal transducer and activator of transcription 5 (STAT5) on α_S1_-casein in goat mammary epithelial cells (GMEC). Deletion analysis showed that the core promoter region of *CSN1S1* was located at −110 to −18 bp upstream of transcription start site, which contained two putative STAT5 binding sites (gamma-interferon activation site, GAS). Overexpression of *STAT5a* gene upregulated the mRNA level and the promoter activity of the *CSN1S1* gene, and STAT5 inhibitor decreased phosphorylated STAT5 in the nucleus and *CSN1S1* transcription activity. Further, GAS site-directed mutagenesis and chromatin immunoprecipitation (ChIP) assays revealed that GAS1 and GAS2 sites in the *CSN1S1* promoter core region were binding sites of STAT5. Taken together, STAT5 directly regulates *CSN1S1* transcription by GAS1 and GAS2 sites in GMEC, and the mutation of STAT5 binding sites could downregulate *CSN1S1* expression and decrease α_S1_-casein synthesis, which provide the novel strategy for reducing the allergic potential of goat milk and improving milk quality in ruminants.

## 1. Introduction

Dairy products are rich in nutrients such as essential amino acids and bioactive peptides needed by human body, which are important sources of dietary protein. Milk proteins are divided into caseins and whey proteins [1]. Caseins consist of α_S1_-, α_S2_-, β- and κ-casein, and whey proteins mainly include α-lactalbumin and β-lactoglobulin [2]. Despite the nutritiousness of milk proteins, they also carry the risk of a milk allergy for many people [3]. Milk protein contains antigen epitopes that can be recognized by the human immune system, and may cause skin itching, vomiting, diarrhea, allergic rhinitis and other adverse symptoms [4]. The prevalence of a food allergy caused by milk protein is as high as 2–7.5% in different countries, especially for infants and children [5,6]. Among various milk protein components, α_S1_-casein is highly susceptible to human allergies, which is sensitive for approximately 65% of patients suffering from a milk protein allergy [7,8]. Goat milk contains a considerable variation of α_S1_-casein content (4–26% of total protein) compared to cow milk (36–40%) [9,10]. Therefore, the means of manipulating α_S1_-casein gene transcription to decrease milk allergic reactions is necessary to be elucidated.

Transcription factors that play key roles in regulating the transcription of milk protein genes mainly include signal transducer and activator of transcription 5 (STAT5), glucocorticoid receptor (GR), activator protein 1 (AP-1), CCAAT enhancer binding protein (C/EBP) and YinYang1 (YY1) in mammals. Among these transcription factors, STAT5 plays a central role in the transcription of milk protein genes [11]. During the lactation period, hormones such as prolactin, glucocorticoids and insulin could bind to corresponding receptors on the mammary epithelial cell membranes to activate STAT5 [12]. Then phosphorylated STAT5 transfers to the nucleus and binds specifically to the GAS (gamma-interferon activation site) response element of the milk protein gene promoter regions to initiate the transcription process [13,14]. Breast specific STAT5-knockout mice had impaired development of the mammary gland and the decrease of whey protein content in milk [15]. In the mammary epithelial cell lines of cows [16] and dairy goats [17], STAT5 binds to the *CSN2* gene promoter and promotes the transcription and synthesis of β-casein.

To address the situation of milk allergic reactions due to α_S1_-casein in ruminants, it is essential to clarify the transcriptional regulation of α_S1_-casein (*CSN1S1*). Although previous studies have shown that STAT5 plays crucial roles in the processes of milk protein synthesis; however, whether it could regulate α_S1_-casein transcription in ruminants remains obscure. Therefore, the objective of the present study was to dissect the transcriptional regulation mechanism of *CSN1S1* by STAT5 in goat mammary epithelial cells (GMEC). Here, we found that STAT5 directly regulated the transcription of the *CSN1S1* gene by GAS binding sites in GMEC, which provides the novel way to reduce milk allergic reactions through the mutation of these GAS sites in ruminants.

## 2. Materials and Methods

### 2.1. Ethics Statement

All experimental procedures were approved by the Institutional Animal Care and Use Committee of Northwest A&F University, YangLing, China (protocol number 15-516).

### 2.2. Goat CSN1S1 Promoter Region Cloning and Bioinformatics Analysis

Goat genome sequence of the *CSN1S1* promoter region (GenBank no. NC_030813.1) was used for primer design. Genomic DNA extracted from goat blood samples by using TIANamp Genomic DNA Kit (DP304, Tiangen, Beijing, China). The 2201 bp fragment of the *CSN1S1* gene promoter was amplified and ligated to pMD-19T vector (Takara, Shiga, Japan). The sequences of forward −2018 and reverse +183 primers were shown in Table 1. Then, JASPAR (http://jaspar.genereg.net/, accessed on 20 April 2020) and TRANSFAC databases (http://www.gene-regulation.com, accessed on 20 April 2020) were used to predict the putative binding sites of the transcription factor in the *CSN1S1* promoter region.

### 2.3. Vector Construction

The deletion fragment primers of the *CSN1S1* promoter for −1715, −1354, −1049, −598, −346, −110 and −18 bp were designed (Table 1), then amplified using −2018/+183 bp as a template. DNA fragments of the *CSN1S1* promoter region were gel purified, digested with *KpnI* and *HindIII* (Takara) and cloned into a pGL3-Basic vector. The plasmids were confirmed by gel electrophoresis (Appendix A) and sequencing. The site-directed mutant primers of STAT5 binding sites for GAS1, GAS2 and GAS3 in the *CSN1S1* promoter were showed in Table 1. Overlapping PCR technology was used to construct GAS site-directed mutagenesis vectors. All plasmids were verified by DNA sequencing.

### 2.4. Cell Culture

The GMEC were isolated from 3 healthy Xinong Saanen dairy goats at the peak of lactation, as previously described [18]. The protocol for the purification and identification procedure have been described in the previous study [19]. Cells were investigated by immunofluorescence with Cytokeratin 18 (66187-1-Ig, Proteintech, Rosemont, IL, USA, 1:200; Appendix A), which is specific for epithelial cells. Cells were cultured at 37 °C in 5% CO_2_. The culture medium contained basal DMEM/F12 (SH30023, Hyclone, Logan, UT, USA) medium, 5 mg/L insulin (16634, Sigma, St. Louis, MO, USA), 1 mg/L hydrocortisone (H0888, Sigma), 10 μg/L epidermal growth factor (PHG0311, Invitrogen, Carlsbad, CA, USA), 1 × 10^5^ U/L streptomycin/penicillin (080092569, Harbin Pharmaceutical Group, Harbin, China) and 10% FBS (10099141, Gibco, Gaithersburg, MD, USA). To promote lactogenesis, GMEC were switched to a lactogenic medium supplemented with 2 mg/L prolactin (L6520, Sigma) for 48 h before performing the experiments.

### 2.5. Cell Treatment

The GMEC were approximately 80% confluence before applying treatments. For luciferase assays, cells were cultured in 48-well plates at a density of 5 × 10^4^ cells per well and transfected with the *CSN1S1* promoter vectors by X-treme GENE HP DNA Transfection Regent (06366236001, Roche, Mannheim, Germany) for 48 h. Similarly, cells were incubated with pcDNA3.1-STAT5a expression vector [20] or STAT5 inhibitor STAT5-IN-1 (CAS 285986-31-4, MedChemExpress, Monmouth Junction, NJ, USA; 200 μM, dissolved in DMSO) followed by *CSN1S1* promoter plasmids transfection, respectively. The pRL-TK vector (renilla luciferase) was the internal control of pGL3-Basic vector (the mass ratio was 1:50). For quantitative real-time PCR (qRT-PCR) assays, cells were cultured in 12-well plates at a density of 2 × 10^5^ cells per well and treated with pcDNA3.1-STAT5a vector or STAT5-IN-1 (200 μM) for 48 h, respectively. For western blot analysis, cells were cultured in 60 mm dish and incubated with STAT5-IN-1 (200 μM) for 48 h, then cellular nuclear and cytoplasmic proteins were extracted, respectively. In addition, cells seeded in the culture medium were incubated with STAT5-IN-1 (200 μM, or DMSO) followed by prolactin (2 mg/L) treatment for 48 h, then total RNA or protein was extracted. Similarly, cells seeded in the culture medium were co-treated with pGL3-CSN1S1 vector and prolactin (2 mg/L) for luciferase assays.

### 2.6. Luciferase Assays

The GMEC cultured in 48-well plates were harvested and lysed for luciferase activity measurement. The luciferase activity was detected by the Dual-Luciferase Reporter System (E1910, Promega, Madison, WI, USA) on a Fluoroskan Ascent (ThermoFisher Scientific, Waltham, MA, USA). The relative luciferase activity was calculated as the ratio of firefly compared with renilla luciferase activity.

### 2.7. RNA Extraction and Quantitative Real-Time PCR

The total RNA of GMEC was extracted by Trizol reagent (9109, Takara). The concentration and purity of RNA were measured by NanoDrop2000 (ThermoFisher Scientific). The optical density 260/280 was 1.9–2.1, and the concentration of RNA was 400–500 ng/μL. The purified RNA was used to synthesize cDNA by PrimeScript RT kit (RR047A, Takara). qRT-PCR were detected by SYBR green (RR820A, Takara) in a CFX96 sequence detector (Bio-Rad, Hercules, CA, USA). The qRT-PCR reaction condition was performed at 95 °C for 30 s, followed by 40 cycles of 95 °C for 5 s and 60 °C for 30 s. Ribosomal protein S9 (*RPS9*) and ubiquitously expressed transcript (*UXT*) were used as internal control genes [21], and data were analyzed using 2^−ΔΔCt^ method [22]. Primers for qRT-PCR are shown in Table 2.

### 2.8. Western Blot

Total protein of GMEC was harvested using RIPA lysis buffer (R0010, Solarbio) containing phosphatase and protease inhibitor cocktails (04906845001 and 04693132001, Roche). Nuclear and cytoplasmic proteins were extracted separately according to the protocol of protein extraction kit (DE201, TransGen, Beijing, China). Protein concentration was measured by BCA kit (23227, Thermo Scientific). The protein was separated using 10% SDS-PAGE and blotted onto a PVDF membrane (Roche) by a Bio-Rad Trans-Blot SD. Nonspecific binding sites were blocked for 2 h by 5% skim milk (232100, BD Biosciences, Franklin Lakes, NJ, USA), then probed overnight at 4 °C with primary antibodies: rabbit anti-α_S1_-casein (SAB1401093, Sigma-Aldrich; 1:1000), mouse anti-p-STAT5a (611964, BD Biosciences; 1:500), mouse anti-STAT5a (610191, BD Biosciences; 1:1000), mouse anti-Histone H3 (100005-MM01, Sino Biological, Beijing, China; 1:2000) and mouse anti-β-actin (CW0096, CW Biotech; 1:2000). Horseradish peroxidase-conjugated goat anti-mouse-IgG (CW0102, CW Biotech; Beijing, China; 1:5000) and goat anti-rabbit-IgG (CW0103, CW Biotech; 1:5000) secondary antibodies were performed. Signals were examined by the chemiluminescent kit (1705061, Bio-Rad). ImageJ (http://imagej.nih.gov, accessed on 16 November 2020) was used for measuring the intensities of bands. Relative expression of α_S1_-casein was normalized to β-actin, and p-STAT5a expression was normalized by comparison to *STAT5a* or Histone H3.

### 2.9. Chromatin Immunoprecipitation (ChIP) Assays

The ChIP assays were performed according to the protocol of a commercial kit (P2048, Beyotime, Shanghai, China). GMEC were cultured in 100 mm dish at a density of 2 × 10^6^ cells per well until 80% confluence before cell collection. Cells were crosslinked with a final concentration of 1% formaldehyde for 10 min at 37 °C, then washed in cold PBS containing phosphatase and protease inhibitors. Chromosomal DNA was sheared into lengths of 100 to 500 bp by Bioruptor UCD-200 (Diagenode, Seraing, Belgium). The chromatin–protein complexes were treated with protein A/G beads and rabbit p-STAT5 antibody (9351S, Cell Signaling Technology, Danvers, MA, USA; 1:50) or normal rabbit IgG antibody (A7016, Beyotime; 1:50) at 4 °C for 12 h. The primers used in the ChIP assays for amplification are shown in Table 2.

### 2.10. Statistical Analysis

All experiments were carried out with each goat as a biological replicate (*n* = 3). The data were processed using SPSS20.0 (IBM, Chicago, IL, USA). Statistical analysis was performed using Student’s t-test for two group comparison, and one-way ANOVA was performed with Tukey test for multiple comparisons. Data were shown as means ± SEM. Significant differences were declared at *p* < 0.05 (* *p* < 0.05, ** *p* < 0.01).

## 3. Results

### 3.1. Cloning and Characterization of the Goat CSN1S1 Promoter

We obtained a 2201-bp 5′-flanking sequence of the *CSN1S1* promoter containing 2018 bp upstream of the transcription start site (+1), the first exon (+1 to +107 bp) and part of the first intron. Using the JASPAR and TRANSFAC database analysis of the *CSN1S1* promoter sequence, we revealed some consensus binding sites of transcription factors, such as STAT5 (−46, −92, −1345), AP-1 (−131, −566), C/EBP (−318, −420, −459), GR (−228, −306), YY1 (−206, −341) and TATA-box (−33, Figure 1).

### 3.2. Core Promoter Region Identification of CSN1S1 Gene

The pGL3-CSN1S1 vector was transfected into GMEC, and luciferase activity was measured. The result shows that the *CSN1S1* promoter activity is more active than the control vector (*p* < 0.01, Figure 2A). Eight promoter fragments of various lengths (−2018/+183, −1715/+183, −1354/+183, −1049/+183, −598/+183, −346/+183, −110/+183, −18/+183 bp) were cloned into pGL3-Basic vectors to confirm the core region of transcription activity in the *CSN1S1* promoter. The luciferase assays showed that there are transcriptional activator elements in −2018~−1715 bp and −1354~−1049 bp regions, while transcriptional suppressor elements are present in −1715~−1354 bp and −1049~−598 bp regions. The promoter activity was progressively declined when the region between −598 and −110 bp slightly deleted (*p* < 0.05; Figure 2B). Further deletion from −110/+183 to −18/+183 bp reduced the activity by 89%, almost abolishing the promoter activity (*p* < 0.05; Figure 2B). These findings suggest that the region from −110 to −18 bp is required for maintaining basal promoter activity of the goat *CSN1S1* gene, which contains two putative STAT5 binding sites (GAS sites).

### 3.3. STAT5 Upregulates CSN1S1 Expression

Next, to define the regulator effect of STAT5 on α_S1_-casein transcription, GMEC were treated with pcDNA3.1-STAT5a or STAT5-IN-1. Compared with control, mRNA abundance of *STAT5a* and *CSN1S1* was effectively increased in *STAT5a* overexpression group (*p* < 0.01, Figure 3A); moreover, the *CSN1S1* promoter activity was also enhanced (*p* < 0.01, Figure 3B). Compared with the DMSO group, STAT5 inhibitor STAT5-IN-1 decreased the mRNA level and the promoter activity of *CSN1S1* (*p* < 0.01, Figure 3C,D). In addition, *STAT5a* phosphorylation in nuclear was markedly reduced (*p* < 0.05), as well as α_S1_-casein synthesis in cytoplasm by STAT5 inhibition (*p* < 0.01, Figure 3E). Collectively, we indicate that STAT5 activates *CSN1S1* transcription activity, and *CSN1S1* may be a target gene of STAT5 in GMEC.

### 3.4. Involvement of STAT5 in Transcriptional Regulation of CSN1S1

Bioinformatics analysis revealed three conserved GAS binding sites in the *CSN1S1* promoter region (Figure 1). To examine which elements are functional for STAT5-mediated *CSN1S1* regulation, vectors containing GAS site-mutated (GAS1, GAS2 and GAS3) versions of the *CSN1S1* promoter were transfected into GMEC for promoter activity assays. Compared with the wild-type promoter vector, GAS1 or GAS2 mutation decreased the *CSN1S1* promoter activity by an average 40% and 48%, respectively (*p* < 0.05; Figure 4). In contrast, individual mutation of GAS3 could not affect the *CSN1S1* promoter activity. Similarly, mutations of both GAS1 and GAS2 sites (or all 3 sites) resulted in very low promoter activity (Figure 4). These results show that the GAS1 and GAS2 binding sites of STAT5 are critical for *CSN1S1* promoter activation.

Further, we overexpressed or inhibited STAT5 to identify the role of GAS1 and GAS2 elements in the regulation of *CSN1S1* by STAT5. The GMEC were co-treated with pcDNA3.1-STAT5a (or STAT5-IN-1) and the site-mutated vectors. Results showed that *STAT5a* overexpression could also increase the *CSN1S1* promoter activity during the individual GAS1 or GAS2 site mutation compared with the pcDNA3.1-NC control (*p* < 0.01, Figure 5A). However, double mutations of GAS1 and GAS2 sites could abolish the stimulatory effect of STAT5 on the *CSN1S1* promoter activity (Figure 5A). Consistently, STAT5 inhibition markedly decreased the *CSN1S1* promoter activity during the individual GAS1 or GAS2 mutation (*p* < 0.05; Figure 5B). While STAT5 inhibition could not affect the promoter activity during the double mutations of GAS1 and GAS2 binding sites (Figure 5B). Collectively, these results indicate that GAS1 and GAS2 are required for *CSN1S1* promoter transcription in response to STAT5.

### 3.5. Prolactin Increases CSN1S1 Promoter Activity via STAT5

In mammary epithelial cells, prolactin could activate STAT5 phosphorylation. To explore whether prolactin affected *CSN1S1* promoter activity and expression via STAT5 signaling, the GMEC seeded in the culture medium were treated with prolactin. Compared with the control, prolactin effectively increased the mRNA level and promoter activity of the *CSN1S1* gene (*p* < 0.05, Figure 6A,B). Additionally, we found that *STAT5a* phosphorylation and α_S1_-casein synthesis were increased after incubation with prolactin (*p* < 0.05, Figure 6C), while STAT5-IN-1 abolished the stimulation of prolactin on p-STAT5a and α_S1_-casein expression (Figure 6C). These findings indicate that the prolactin-mediated increase of *CSN1S1* expression might be involved with STAT5.

Further, we co-treated with GAS site-mutated vectors and prolactin in GMEC to identify the STAT5 binding sites involved in prolactin-induced *CSN1S1* expression. Results showed that the prolactin-mediated increase of *CSN1S1* promoter activity was reduced when a single mutation of GAS1 or GAS2 occurred (*p* < 0.01; Figure 6D). However, the stimulatory effect of prolactin on *CSN1S1* promoter activity was completely abolished in GMEC transfected with the GAS1 and GAS2 simultaneously mutated *CSN1S1* promoter (Figure 6D). Taken together, these results suggest that GAS1 and GAS2 sites are responsible for the activation of prolactin on the activity of the *CSN1S1* promoter.

### 3.6. STAT5 Binds to the GAS Sites of CSN1S1 Promoter Region

To confirm whether the transcription factor STAT5 binds to the GAS1 and GAS2 sites on the *CSN1S1* promoter region, ChIP assays were performed. We found that STAT5 interacted with the *CSN1S1* promoter at the GAS1 and GAS2 sites (Figure 7A,C). qRT-PCR results showed that the fold enrichment levels of GAS1 and GAS2 sites were ~5.9 and 7.3, respectively (Figure 7B,D). Collectively, we conclude that STAT5 directly binds to GAS sites in the *CSN1S1* promoter region, and that the GAS1 and GAS2 sites are necessary for STAT5 to regulate *CSN1S1* transcription and α_S1_-casein synthesis.

## 4. Discussion

As one of the main components of milk protein, α_S1_-casein content is closely related to the milk allergy potential for humans [23]. The problem of α_S1_-casein allergy could be caused by cow milk, goat milk and other ruminant milk [4]. Reducing the α_S1_-casein content in goat milk and other ruminant milk is beneficial for the intake of dairy products for people with a milk protein allergy. The transcription of α_S1_-casein mainly depends on the activity of the *CSN1S1* gene promoter in mammary epithelial cells [24]. Bioinformatics analysis revealed multiple polymorphism sites in the promoter region of the *CSN1S1* gene [25,26]. According to the transcription factor binding sites in the *CSN1S1* gene promoter, the locus mutation of the *CSN1S1* promoter is associated with the transcription efficiency and the α_S1_-casein content in cow milk [27]. There are polymorphism and insertion–deletion sites in the promoter region of the *CSN1S1* gene, which affect the protein content in mare milk [28]. Although the promoter sequence of the *CSN1S1* gene has been well analyzed in silico, the research on its transcriptional regulation mechanism remains scarce [29]. In this study, we focus on the regulation mechanism of *CSN1S1* promoter activity in goat mammary epithelial cells, which contributes to the improvement of milk quality for human consumption [30].

There are many transcription factor binding motifs in the *CSN1S1* promoter region of dairy cows and goats, such as activating factor STAT5, GR, C/EBP, AP-1 and inhibitory factor YY1 [31,32]. The methylation of the milk protein gene promoter affects the binding of transcription factors to cis-acting elements and inhibits the expression of the milk protein [33]. During the lactation period of dairy cows, the methylation level of the specific site in the *CSN1S1* promoter was negatively correlated with the *CSN1S1* mRNA level in the mammary gland [34]. CpG island methylation of the STAT5 binding site in the *CSN1S1* promoter led to the decrease in milk protein production in dairy cows [35]. In mammary epithelial cells, STAT5, GR and C/EBP could interact with acetyltransferase p300 and recruit RNA polymerase II, which affects β-casein gene transcription [36]. In the present study, we cloned 2201 bp sequence of the 5′-flanking promoter region of the goat *CSN1S1* gene. In addition to the TATA-box near the transcription start site, the putative binding sites of transcription factors were comprehensively predicted by using JASPAR and TRANSFAC databases [37]. These transcription factors associated with milk protein synthesis [38], such as STAT5, may modulate the transcriptional activity of the goat *CSN1S1* promoter.

In ruminants, the active center of gene promoter plays a vital role in transcription activity [39,40]. In order to determine the core promoter region of the *CSN1S1* gene in dairy goats, we performed a segment-by-step deletion analysis starting from −2018 bp upstream of transcription start site [41]. When the *CSN1S1* promoter fragment was deleted from −110 to −18 bp, the promoter activity decreased by 89%; therefore, −110~−18 bp upstream of transcription start site was identified as the active center of the goat *CSN1S1* promoter [42]. The core promoter region contains two STAT5 binding sites (GAS1 and GAS2), thus we hypothesized that STAT5 may have a great effect on the transcriptional activity of the *CSN1S1* gene. In addition, another potential STAT5 binding site (GAS3) may also exist in −1354~−1049 bp upstream. Sequence analysis of the *CSN1S1* promoter region in bovine showed that there were three conserved STAT5 binding sites similar to the sequences of dairy goats [32]. Therefore, the investigation of *CSN1S1* promoter activity in dairy goats could also provide the theoretical support for *CSN1S1* transcriptional regulation in bovine, which contribute to the reduction of α_S1_-casein content in ruminant milk.

Previous studies have shown that STAT5 plays the crucial regulatory role in mammalian milk protein synthesis [43]. After the knockout of STAT5 in the mouse mammary gland, the content of α-lactalbumin and β-lactoglobulin in milk was reduced by 23% and 39%, respectively [15]. In addition to STAT5, mammalian target of rapamycin (mTOR) also regulates the synthesis of milk proteins such as α-lactalbumin and α_S1_-casein [44,45]. In bovine mammary epithelial cells, suppressor of cytokine signaling 3 inhibited *STAT5a* activity, and milk protein synthesis was significantly reduced after the phosphorylation of *STAT5a* was blocked [46,47]. In both mouse cell lines and immortalized GMEC, *STAT5a* binds to *CSN2* promoter and promotes the transcription of β-casein [48]. In primary GMEC, *STAT5a* binds to the *CSN2* promoter region, and STAT5 activity inhibition markedly reduced *CSN2* promoter activity and β-casein synthesis [20]. Consistent with the previous studies, in the present study, the inhibition of STAT5 activity significantly reduced phosphorylated *STAT5a* in the nucleus, which functions as a transcription factor, and downregulated *CSN1S1* mRNA level and α_S1_-casein synthesis. Site-directed mutagenesis of the three potential STAT5 binding sites in the *CSN1S1* promoter region revealed that GAS1 and GAS2 sites in the core region could affect the promoter activity, while the distal GAS3 site had no significant effect on the promoter activity. Simultaneous mutation analysis of multiple sites showed that GAS1 and GAS2 sites could independently affect the promoter activity. The activity of the *CSN1S1* promoter was severely weakened after simultaneous mutation of the two sites in the core region, and even the overexpression of *STAT5a* could not enhance its activity.

Prolactin is a key hormone regulating milk protein synthesis in mammals [49]. In bovine mammary epithelial cells, prolactin activates STAT5 phosphorylation via the prolactin receptor, promotes the expression of β-casein gene and maintains cell proliferation and milk protein synthesis [50,51]. Mouse mammary epithelial cells could not synthesize and secrete β-casein properly under the condition of prolactin deficiency, and prolactin induces β-casein synthesis by activating the proximal promoter and distal enhancer of the *CSN2* gene [52]. Therefore, our study attempted to use prolactin to activate the phosphorylation of STAT5 in GMEC. Consistent with the results of direct inhibition of STAT5 activity, the promoter activity of the *CSN1S1* gene was severely inhibited after the mutation of GAS1 and GAS2 sites. In this case, prolactin could also not activate its transcriptional activity, indicating that these two sites were the main binding sites of STAT5 in the *CSN1S1* promoter. Compared with the site mutation analysis, the ChIP assay is the direct evidence to confirm the binding of transcription factors to the promoter region [40,42]. In the present study, ChIP assays showed that STAT5 could directly bind to GAS1 and GAS2 sites and promote the transcription of the *CSN1S1* gene. These results indicate that STAT5 can directly regulate the transcription of the *CSN1S1* gene and the synthesis of α_S1_-casein in GMEC. In addition, the complicated mechanism under which STAT5 binding site mutation at goat genome affects the synthesis of α_S1_-casein and other milk proteins requires further investigation.

## 5. Conclusions

The core promoter region of the goat *CSN1S1* gene is located at −110~−18 bp upstream of transcription start site, which contains two STAT5 binding sites (GAS). STAT5 activates *CSN1S1* gene promoter activity and α_S1_-casein synthesis. Furthermore, GAS1 and GAS2 sites in the core region of the *CSN1S1* promoter are the binding sites of STAT5. Overall, this study reveals that STAT5 can directly regulate *CSN1S1* transcription and α_S1_-casein synthesis through GAS1 and GAS2 sites. Our findings provide the novel basis for decreasing milk allergic reactions by mutating the STAT5 binding sites on α_S1_-casein gene promoter in ruminants.

## Figures and Tables

**Figure 1 foods-11-00346-f001:**
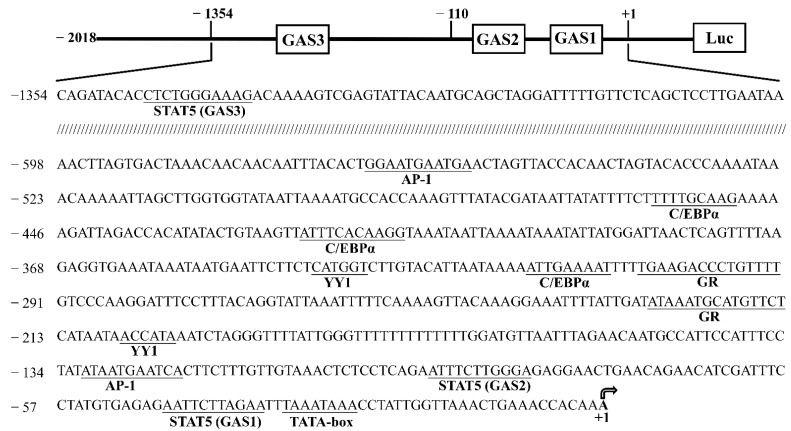
Schematic representation of the goat α_S1_-casein gene (*CSN1S1*) promoter region. The underlined sequence of bases indicates the putative binding sites, and names of transcription factors appear below the underline. +1 represents transcriptional start site and gamma-interferon activation site (GAS) represents the binding site of signal transducer and activator of transcription 5 (STAT5).

**Figure 2 foods-11-00346-f002:**
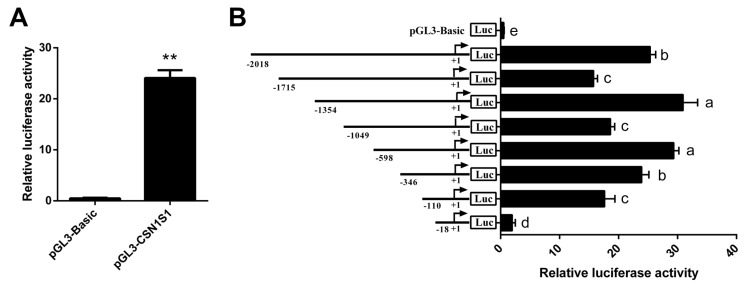
Deletion analysis of the goat *CSN1S1* promoter. (**A**) Relative luciferase activity of full-length *CSN1S1* promoter. (**B**) Relative luciferase activity of the *CSN1S1* promoter in different lengths. Serial *CSN1S1* promoter deletions (−2018/+183, −1715/+183, −1354/+183, −1049/+183, −598/+183, −346/+183, −110/+183, −18/+183 bp) were transfected into GMEC and incubated at 48 h for luciferase assays. Data are shown as means ± SEM for 3 biological replicates. ** *p* < 0.01. Differences between 2 groups are considered significant at lowercase letters, *p* < 0.05.

**Figure 3 foods-11-00346-f003:**
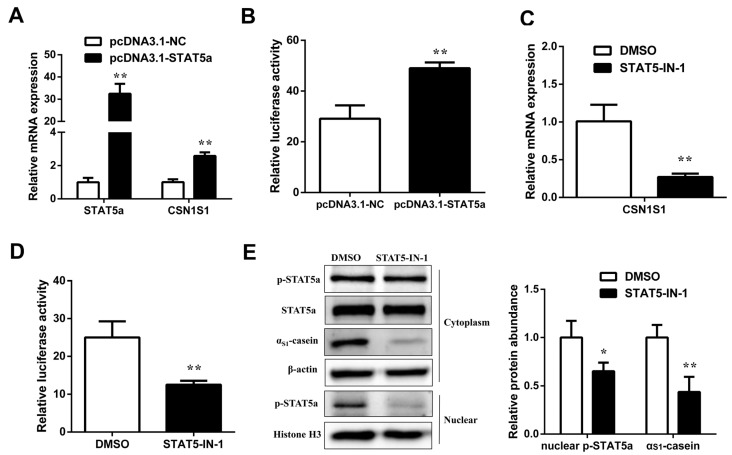
Effects of STAT5 on the expression and the promoter activity of the *CSN1S1* gene in goat mammary epithelial cells (GMEC). (**A**) The mRNA expression levels of *STAT5a* and *CSN1S1* and (**B**) the promoter activity of *CSN1S1* in GMEC transfected with pcDNA3.1-STAT5a or pcDNA3.1-NC for 48 h. (**C**) The mRNA expression level and (**D**) the promoter activity of *CSN1S1* in GMEC treated with STAT5 inhibitor STAT5-IN-1 (200 μM) or DMSO for 48 h. (**E**) GMEC were treated with STAT5-IN-1 (200 μM) or DMSO. After 48 h incubation, p-STAT5a and α_S1_-casein expression were examined. Relative expression of α_S1_-casein was normalized by comparison to β-actin. Relative expression of nuclear p-STAT5a was normalized by comparison to Histone H3. Data are shown as means ± SEM for 3 biological replicates. * *p* < 0.05, ** *p* < 0.01.

**Figure 4 foods-11-00346-f004:**
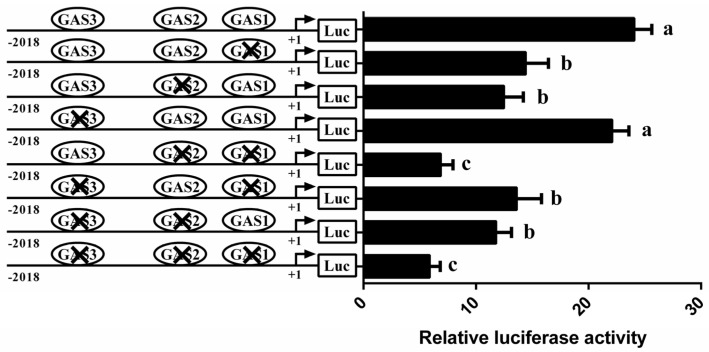
Relative luciferase activities of mutagenesis of STAT5 binding sites (GAS) in the *CSN1S1* promoter region. Data are shown as means ± SEM for 3 biological replicates. Differences between 2 groups are considered significant at lowercase letters, *p* < 0.05.

**Figure 5 foods-11-00346-f005:**
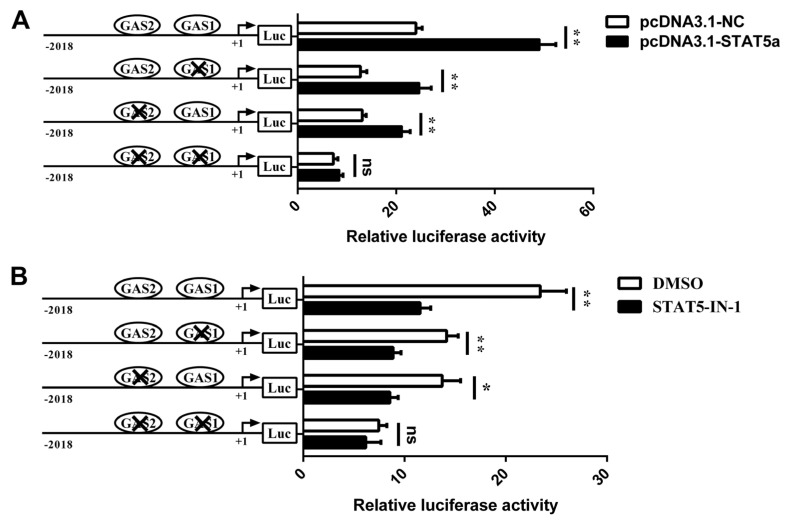
STAT5 regulates the *CSN1S1* promoter activity by directly binding to the GAS sites in the *CSN1S1* promoter region. (**A**) Effects of *STAT5a* overexpression on site-mutations of *CSN1S1* promoter activity. GMEC were transfected with the wild-type pGL3-CSN1S1 (or GAS site-mutated constructs) followed by pcDNA3.1-STAT5a (or pcDNA3.1-NC) transfection, respectively. At 48 h treatment, cells were measured for luciferase activity. (**B**) Effects of STAT5 inhibition on site-mutations of *CSN1S1* promoter activity. GMEC were treated with STAT5-IN-1 (200 μM, or DMSO) before wild-type pGL3-CSN1S1 (or GAS site-mutated constructs) transfection, respectively. At 48 h treatment, cells were measured for luciferase activity. Data are shown as means ± SEM for 3 biological replicates. * *p* < 0.05, ** *p* < 0.01. ns, no significance.

**Figure 6 foods-11-00346-f006:**
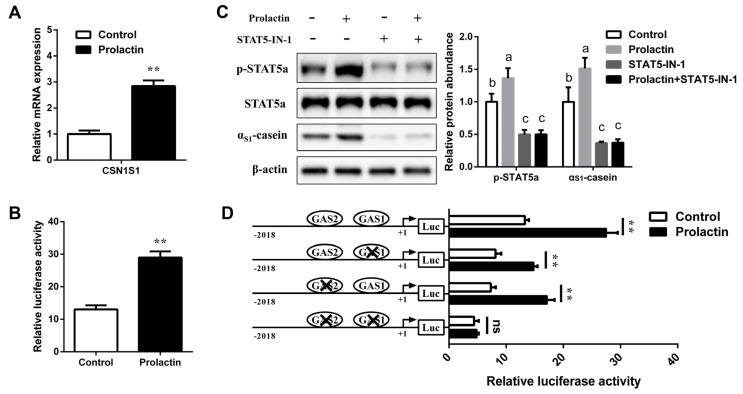
Effects of prolactin on the expression and promoter activity of the *CSN1S1* gene. (**A**) The mRNA expression level and (**B**) promoter activity of *CSN1S1* in GMEC incubated with prolactin (2 mg/L) for 48 h in culture medium. (**C**) GMEC seeded in culture medium were treated with STAT5-IN-1 (200 μM, or DMSO) before prolactin (2 mg/L) treatment for 48 h, then total protein was extracted. Relative expression of α_S1_-casein was normalized by comparison to β-actin. Relative expression of p-STAT5a was normalized by comparison to STAT5a. (**D**) Relative luciferase activity of *CSN1S1* promoter activity in GMEC co-treated with prolactin (2 mg/L) and the wild-type pGL3-CSN1S1 (or GAS site-mutated constructs) for 48 h in culture medium. Data are shown as means ± SEM for 3 biological replicates. ** *p* < 0.01. Differences between 2 groups are considered significant at lowercase letters, *p* < 0.05. ns, no significance.

**Figure 7 foods-11-00346-f007:**
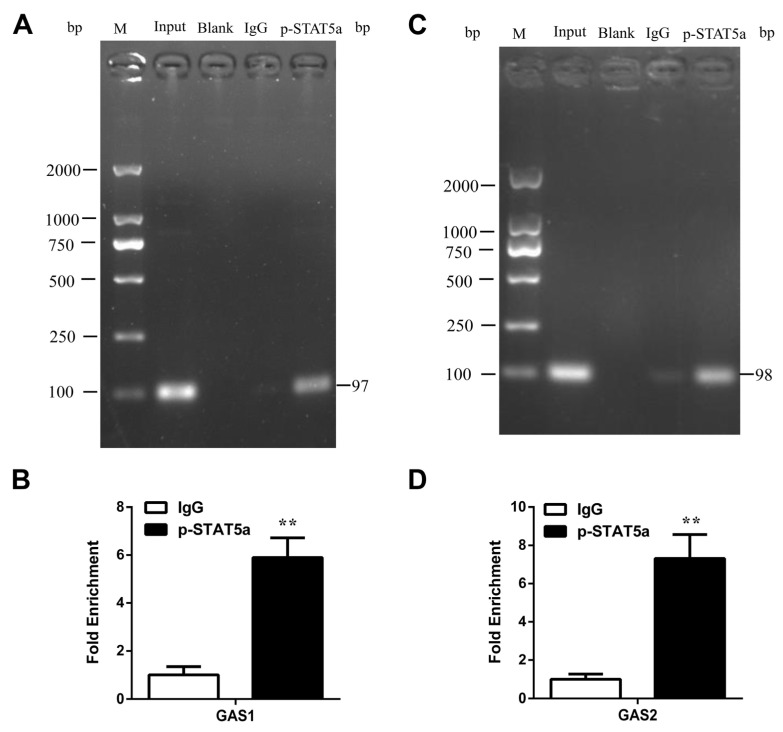
Transcription of the *CSN1S1* gene is regulated by direct STAT5 binding to GAS sites in GMEC. (**A**) GAS1 and (**C**) GAS2 binding sites in *CSN1S1* promoter region were recognized by STAT5 directly using chromatin immunoprecipitation (ChIP)-PCR analysis. (**B**) GAS1 and (**D**) GAS2 binding sites in the *CSN1S1* promoter region were recognized by STAT5 directly using ChIP-qPCR analysis. Data are shown as means ± SEM for 3 biological replicates. ** *p* < 0.01.

**Table 1 foods-11-00346-t001:** Primers used for cloning, deletion and site-directed mutagenesis of the goat *CSN1S1* gene.

Primers	Primer Sequence ^†^ (5′–3′)	Binding Region
Cloning and 5′ deletion primers		
Forward 1	CGGGGTACCGGTGGCAAGAATAGTATTACAG	−2018
Forward 2	CGGGGTACCGACATCACTTTACTGATAAG	−1715
Forward 3	CGGGGTACCCAGATACACCTCTGGGAAAG	−1354
Forward 4	CGGGGTACCGATTCCTTTCTTATAAACAA	−1049
Forward 5	CGGGGTACCAACTTAGTGACTAAACAACA	−598
Forward 6	CGGGGTACCCTTCTCATGGTCTTGTACAT	−346
Forward 7	CGGGGTACCGTAAACTCTCCTCAGAATTTC	−110
Forward 8	CGGGGTACCGTTAAACTGAAACCACAAAAT	−18
Reverse	CCCAAGCTTCAACTGCGTATTAGTGAAGA	+183
Site-directed mutagenesis primers ^‡^		
GAS1-mut	CCTATGTGAGAGAAcgtTTAtgcTTTAAA	−46
GAS1-anti-mut	TAGGTTTATTTAAAgcaTAAacgTTCTCT	−46
GAS2-mut	ACTCTCCTCAGAATcgtTTGtgcGAGGAA	−92
GAS2-anti-mut	CTGTTCAGTTCCTCgcaCAAacgATTCTG	−92
GAS3-mut	AAGTCCAGATACACgcaTGGacgAGACAAAA	−1345
GAS3-anti-mut	TACTCGACTTTTGTCTcgtCCAtgcGTGTAT	−1345

^†^ Lowercase letters represent mutation sites, italics indicate restriction enzyme sites. ^‡^ Mut = mutagenesis forward primer; anti-mut = mutagenesis reverse primer.

**Table 2 foods-11-00346-t002:** Primers used for quantitative real-time PCR.

GenBank ID	Gene	Primer Sequence (5′–3′)	Length (bp)
XM_018049127.1	*CSN1S1*	F, TCCACTAGGCACACAATACACTGA	61
		R, GCCAATGGGATTAGGGATGTC	
XM_018065112.1	*STAT5a*	F, CCATCGACCTGGACAATCCC	96
		R, CGACTTGGTGCTCTGCCTTCTT	
XM_005700842.2	*UXT*	F, CAGCTGGCCAAATACCTTCAA	125
		R, GTGTCTGGGACCACTGTGTCAA	
XM_005709411.1	*RPS9*	F, CCTCGACCAAGAGCTGAAG	64
		R, CCTCCAGACCTCACGTTTGTTC	
ChIP primers	GAS1	F, GAGGAACTGAACAGAACATC	97
		R, GTAAAATGCTAATTTTGTGG	
	GAS2	F, ACAATGCCATTCCATTTCCTAT	98
		R, AGGAAATCGATGTTCTGTTCAG	

## Data Availability

Not applicable.

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
