# Peer review of "Mutation of Signal Transducer and Activator of Transcription 5 (STAT5) Binding Sites Decreases Milk Allergen αS1-Casein Content in Goat Mammary Epithelial Cells"

_foods, 2022, doi:10.3390/foods11030346_

Round 1

Reviewer 1 Report

Dear Authors,

The evaluation of the study has been completed and my suggestions to the authors are presented below. This study has the potential to make a significant contribution to animal husbandry. 

- Line 30-64: In introduction section, a brief information should be given about the casein proteins in goat's milk in comparison with other farm animals.

- Line 72-73: The method of DNA extraction should be explained.

- Line 126-136: In addition to purity of extracted RNA, concentration values of samples might be presented here.

- Line 135: Reference should be added.

- Line 263: The full name of the abbreviation "NS" should be given.

- Line 290: The full name of the abbreviation "NS" should be given.

- Line 298-299: More clear gel images might be used.

- Line 316-320: Reference/s should be added.

- Reference usage should be preferred for some of the information given in the discussion section.

- The limitations of the study should be mentioned in discussion section.

Author Response

Response to Reviewer 1 Comments

Dear Reviewer 1:

Thanks for your comments. We have made corrections as suggested. Thank you very much.

Point 1: The evaluation of the study has been completed and my suggestions to the authors are presented below. This study has the potential to make a significant contribution to animal husbandry.

- Line 30-64: In introduction section, a brief information should be given about the casein proteins in goat's milk in comparison with other farm animals.

Response 1: Thanks for your comment. Cow milk and goat milk are the two most common dairy products. Goat milk contains considerable variation of αS1-casein content (4-26% of total protein) than cow milk (36-40%), which is a big difference between goat milk and cow milk in casein proteins (Carillier-Jacquin et al., 2016, Clark and Sherbon, 2000). As suggested, we have presented a brief comparison of casein proteins between goat milk and cow milk, please see Line 41-42.

Point 2: - Line 72-73: The method of DNA extraction should be explained.

Response 2: Genomic DNA extracted from goat blood samples by using TIANamp Genomic DNA Kit (DP304, Tiangen, Beijing, China) according to the instructions. As suggested, we have revised the sentence, please see Line 72-73.

Point 3: - Line 126-136: In addition to purity of extracted RNA, concentration values of samples might be presented here.

Response 3: The concentration of RNA samples was 400-500 ng/μL. As suggested, we have presented the concentration values of RNA, please see Line 130.

Point 4: - Line 135: Reference should be added.

Response 4: Reference to the analytical method for quantitative real-time PCR data has been added, please see Line 136 (Reference 22).

Point 5: - Line 263: The full name of the abbreviation "NS" should be given.

Response 5: “ns” means “no significance”. As suggested, we have presented the full name of “ns”, please see Line 261.

Point 6: - Line 290: The full name of the abbreviation "NS" should be given.

Response 6: “ns” means “no significance”. As suggested, we have presented the full name of “ns”, please see Line 288.

Point 7: - Line 298-299: More clear gel images might be used.

Response 7: As suggested, we have presented the clear and full gel images, please see Figure 7A,C in Line 297.

Point 8: - Line 316-320: Reference/s should be added.

Response 8: As suggested, we have added the references (Reference 29 and 30), please see Line 315-318.

Point 9: - Reference usage should be preferred for some of the information given in the discussion section.

Response 9: As suggested, we have added references (Reference 23, 24, 29, 30, 37, 38, 43-45 and 49) for some of the information given in the discussion section, please see Line 305, 309, 316, 318, 331, 332, 349, 352 and 369.

Point 10: - The limitations of the study should be mentioned in discussion section.

Response 10: Based on our study, the effect of STAT5 binding site mutation at goat genome on the synthesis of αS1-casein and other milk proteins requires further investigation. This may be carried out with CRISPR/Cas9 gene editing system in goat mammary epithelial cells. We have presented the limitations of this study, please see Line 385-387.

References:

Carillier-Jacquin, C. and H. Larroque and C. Robert-Granie. 2016. Including a(s1) casein gene information in genomic evaluations of French dairy goats. Genet. Sel. Evol. 48:54.

Clark, S. and J. W. Sherbon. 2000. Genetic variants of alpha(s1)-CN in goat milk: breed distribution and associations with milk composition and coagulation properties. Small Ruminant Res. 38:135-143.

Reviewer 2 Report

The aim of the study was to dissect the transcriptional regulation mechanism of CSN1S1 by signal transducer and activator of transcription 5 (STAT5) in goat mammary epithelial cells. The aim of the research is preceded by Introduction with correctly selected references.

The aim of the research indicates that the manuscript is very valid and significant from the point of view of contemporary diet and nutritional trends.

Materials and methods have been described in a sufficient way, I have no objections to this part of manuscript. Statistical methods are also described in the correct way.

Results are well described and discussed using well-chosen references. I have no objections to this part of manuscript.

However, the discussion of the results should additionally take into account the incomplete impact of STAT5 on the level of α-lactalbumin and β-lactoglobulin in milk (see line 352). Which means that signal transducer and activator of transcription 5 is not the only one to ensure activity and αS1-casein synthesis.

Author Response

Response to Reviewer 2 Comments

Dear Reviewer 2:

Thanks for your comments. We have made corrections as suggested. Thank you very much.

Point 1: The aim of the study was to dissect the transcriptional regulation mechanism of CSN1S1 by signal transducer and activator of transcription 5 (STAT5) in goat mammary epithelial cells. The aim of the research is preceded by Introduction with correctly selected references.

The aim of the research indicates that the manuscript is very valid and significant from the point of view of contemporary diet and nutritional trends.

Materials and methods have been described in a sufficient way, I have no objections to this part of manuscript. Statistical methods are also described in the correct way.

Results are well described and discussed using well-chosen references. I have no objections to this part of manuscript.

However, the discussion of the results should additionally take into account the incomplete impact of STAT5 on the level of α-lactalbumin and β-lactoglobulin in milk (see line 352). Which means that signal transducer and activator of transcription 5 is not the only one to ensure activity and αS1-casein synthesis

Response 1: Thanks for your suggestions. The synthesis of milk proteins is regulated by various factors among which signal transducer and activator of transcription 5 (STAT5) and mammalian target of rapamycin (mTOR) play key roles (Bionaz and Loor, 2011, Castro et al., 2016). In addition to STAT5, mTOR activity also affects the content of α-lactalbumin and αS2-casein in goat milk (Cai et al., 2020); mTOR promotes the synthesis of milk proteins such as αS1-casein in bovine mammary epithelial cells (Luo et al., 2018). We have added the sentence describing the effect of mTOR on milk protein synthesis, please see Line 351-352.

References:

Bionaz, M. and J. J. Loor. 2011. Gene networks driving bovine mammary protein synthesis during the lactation cycle. Bioinformatics and Biology Insights 5:83-98.

Cai, J., D. Wang, F. Zhao, S. Liang, and J. Liu. 2020. AMPK-mTOR pathway is involved in glucose-modulated amino acid sensing and utilization in the mammary glands of lactating goats. Journal of Animal Science and Biotechnology 11:32.

Castro, J. J., S. I. Arriola Apelo, J. A. Appuhamy, and M. D. Hanigan. 2016. Development of a model describing regulation of casein synthesis by the mammalian target of rapamycin (mTOR) signaling pathway in response to insulin, amino acids, and acetate. J. Dairy Sci. 99:6714-6736.

Luo, C., S. Zhao, M. Zhang, Y. Gao, J. Wang, M. D. Hanigan, and N. Zheng. 2018. SESN2 negatively regulates cell proliferation and casein synthesis by inhibition the amino acid-mediated mTORC1 pathway in cow mammary epithelial cells. Sci. Rep.-Uk 8:3912.
